# Simulating the Dynamic Intra-Tumor Heterogeneity and Therapeutic Responses

**DOI:** 10.3390/cancers14071645

**Published:** 2022-03-24

**Authors:** Yongjing Liu, Cong Feng, Yincong Zhou, Xiaotian Shao, Ming Chen

**Affiliations:** 1Department of Bioinformatics, College of Life Sciences, Zhejiang University, Hangzhou 310058, China; coy@zju.edu.cn (Y.L.); ventson@zju.edu.cn (C.F.); yczhou@zju.edu.cn (Y.Z.); 2Biomedical Big Data Center, The First Affiliated Hospital, Zhejiang University School of Medicine, Zhejiang University, Hangzhou 310058, China; 3Joint Research Centre for Engineering Biology, Zhejiang University-University of Edinburgh Institute, Zhejiang University, Haining 314400, China; 3190110633@zju.edu.cn

**Keywords:** tumor heterogeneity, clonal evolution, targeted therapy, treatment outcomes, simulation model

## Abstract

**Simple Summary:**

Various aspects of intra-tumor heterogeneity are key factors for improving clinical practice, but studies on them remain incomplete. Based on the clonal evolution theory, this study proposes a model to simulate the temporal variability of intra-tumor heterogeneity of cancer cell subpopulations. Multiple types of therapies are also incorporated in the model for the simulation of treatment responses at different time points. The simulation results in this study indicate the importance of the timing of therapy and the superiority of neoadjuvant therapy before surgery. The model is incorporated in a webserver for the convenience of understanding the roles of heterogeneity in cancer treatment response.

**Abstract:**

A tumor is a complex tissue comprised of heterogeneous cell subpopulations which exhibit substantial diversity at morphological, genetic and epigenetic levels. Under the selective pressure of cancer therapies, a minor treatment-resistant subpopulation could survive and repopulate. Therefore, the intra-tumor heterogeneity is recognized as a major obstacle to effective treatment. In this paper, we propose a stochastic clonal expansion model to simulate the dynamic evolution of tumor subpopulations and the therapeutic effect at different times during tumor progression. The model is incorporated in the CES webserver, for the convenience of simulation according to initial user input. Based on this model, we investigate the influence of various factors on tumor progression and treatment consequences and present conclusions drawn from observations, highlighting the importance of treatment timing. The model provides an intuitive illustration to deepen the understanding of temporal intra-tumor heterogeneity dynamics and treatment responses, thus helping the improvement of personalized diagnostic and therapeutic strategies.

## 1. Introduction

Tumor heterogeneity has been recognized for a long time [1]. Owing to the tumor hallmark “genomic instability”, mixed populations are believed to exist in most tumors [2]. While all of the tumor cells share similar traits, each of the divergent cell subgroups has its potentially distinct properties. The existence of multiple distinct subpopulations in cancer grants cancer the potential to adapt various conditions and resist therapies, which makes cancer a barely curable malignancy [3]. A variety of therapeutic methods have been developed aiming at the differences between tumor cells and normal cells. For instance, cancer surgery takes advantage of the positional difference in a solid tumor, while radiation therapy exploits the tumor’s higher sensitivity to radiation-induced apoptosis [4]. However, surgeries would be inappropriate for serious invasive or metastatic tumors, non-solid tumors and tumors in vital organs [5]. Radiation therapy may cause severe damage to the surrounding tissues and even give rise to new mutations [6]. Better therapies that would eliminate tumor cells are still being pursued. Compared to normal cells, the most iconic feature of tumor cells is their genomic aberrations. These aberrations lead to a series of alterations, across levels from transcription and translation to metabolism, and each could be a therapeutic target [7]. This makes therapies targeting cancer-specific molecular signatures the most promising therapeutic strategy. A typical example is C*AR*-T [8], which targets the tumor-specific neoantigens on the surface of tumor cells. Hundreds of recent researches have evidenced the excellent performance of C*AR*-T [9,10,11], but half of patients still experienced relapses [12]. Heterogeneity is an important cause of this phenomenon, and its relationship with treatment response remains to be explored. A proper assessment of tumor heterogeneity will offer useful information for the diagnosis and treatment of cancer.

Cancer is considered to be subject to Darwinian evolutionary processes [13]. Tumor cell populations under selective pressure tend to adapt to the environments by increasing fitness, a measurement of the ability to survive and reproduce [14]. In the evolutionary process, advantageous traits are passed on to more offspring, while the individuals with unfavorable traits may not survive and breed, leading to a decreasing proportion of this subpopulation (Figure 1A). According to the clonal evolution theory [15], cells may have accumulated more fitness than normal cells, but are still regulated by governing mechanisms. Some key variants have to be obtained before these precancerous cells start uncontrolled proliferation. These mutations contributing to mutagenesis are called “driver mutations”, while other mutations are considered as by-products unwanted by tumor cells, namely “passenger mutations”. With the accumulation of driver events, cells would gain various beneficial traits (hallmarks of cancers), including increased proliferation, reduced apoptosis or increased ability to remodel the microenvironment [2], and eventually break the homeostasis. In some situations, the comprehensive genomic profiles can be reduced to profiles of a few driver genes. Molecular subtypes are then identified by the expression or mutation patterns, and different subtypes may have divergent prognostic signatures and treatment strategies [16]. Categorizing subtypes is a qualitative measure of inter-tumor heterogeneity. However, corresponding treatment does not always receive expected outcomes due to other minority subpopulations remaining in the tumor mass [17]. In this way, intra-tumor heterogeneity can be deemed as the mixing status of heterogeneous cell subpopulations characterized by driver mutations.

The intra-tumor heterogeneity could be highly dynamic during tumor progression. For most organisms, the time for a significant change of a particular variant’s frequency is often measured in centuries. As for tumors, the same process could be completed in months [18]. The ultra-fast speed of cell generations makes tumors a great reference to study population genetics. Within this perspective, a tumor progression model may have significance beyond its own setting. Simulating tumor progression has always been a central topic of interest, and multiple studies have constructed models with different focuses and underlying theories [19,20,21].

In this paper, we propose a model where various factors are incorporated and reduced to limited attributes. The aim of this model is not to precisely forecast the progression of tumors, but to estimate the trend of subpopulation growth, while providing a simple yet intuitive visualization. With the aid of this model, we have simulated the general patterns of tumor cell proliferation and competition, investigated the therapeutic effects of different heterogeneity levels, time points, time intervals and therapies. The results add some intuitive explanations to tumor recurrence, highlight the importance of intra-tumor heterogeneity in clinic, and provide some new suggestions on treatment timing. This work may shed light on future studies of tumor heterogeneity.

## 2. Materials and Methods

### 2.1. Model Construction

Based on the clonal evolution theory, we construct a deliberately simplified model to simulate tumor progression (Figure 1B). The various factors affecting cell fitness are simplified into two attributes represented by the two parameters, *DR* and *AR* (Table 1). *DR* represents the intensity of proliferation, and *AR* represents the average survival time of the cells. Due to the fact that each cell independently divides, live or decease, a probability ratio is suitable for simulating the tumor growth process. The accumulated Poisson distribution is used to simulate the proliferation of individual tumor cells, while the number of deceased cells is calculated according to the current apoptosis rate *AR*. As the supplement of nutrients and oxygen is limited, the tumor cannot grow indefinitely, so we applied a log constraint correlated to *AR*, *DR* and the current tumor size to simulate a Gompertzian function behavior. Therefore, after the tumor has grown to a certain size, the subpopulations will compete with each other over limited resources. Our model is established for the well-mixed system with no spatial dimension/structure, which is not appropriate for liquid tumors whose spatial characteristics may complicate the competition of different subpopulations.

Equation (1) is used to calculate the tumor size *n* for each subpopulation according to the time *t*. In the simulations, the time step = *T*.
(1)$$\frac{\partial n}{\partial t}=\left(\sum_{1}^{n}P\left(DR\right)-\lfloor n\times AR\rfloor \right)\times e^{\frac{-N\times DR\times AR}{c}}$$

Note: The function *P*(*x*) in the formula returns a random sample from a list of non-negative integers following the Poisson distribution with a mean value *x*. The value *N* is the current total size of all cell subpopulations, the constant *c* is the parameter to limit the total size of a tumor.

Subsequently, to generate different subpopulations, we take the mutation rate into consideration. Various attributes of tumor cells are updated in a stepwise manner at each time interval *T*. For every iteration, each subpopulation has a small chance to harbor passenger mutations, and a smaller chance to harbor a driver mutation, based on the mutation rate *MR*. The “driver mutation” here represents any variant at any level that provides a significant fitness advantage, which might be regarded as a unique pathway-level impact that could be targeted. Cells carrying the new driver mutation are identified as a new subpopulation, and therefore a new subpopulation is assured to have a higher fitness level. The maximum co-existing number of subpopulations is set as 10. These limitations allow a more explicit graphical presentation without losing too much information, as subclones with less fitness can hardly compete with other subclones and grow to a noteworthy amount. Driver mutations in the model are roughly divided into three categories, representing different groups of hallmarks. Mutations belonging to different groups may bring global fitness for all *AR*, *DR*, and *MR*, but emphasizing different attributes. For example, a mutation activating the “Proliferative signaling” hallmark will lead to a high increase in *DR*, a scarce decrease in *AR*, and a mild increase in *MR*. Passenger mutation events also have random but insignificant impacts on all of the attributes. We use Equations (2) and (3) to simulate the alterations introduced by passenger mutations for each time step T.
(2)$$\begin{array}{c} \frac{\partial DR}{\partial t}=\sum_{0}^{\lfloor MR*R\left(0,z\rfloor \right)}R\left(x_{1},y_{1}\right) \end{array}$$
(3)$$\begin{array}{c} \frac{\partial AR}{\partial t}=\left(\sum_{0}^{\lfloor MR*R\left(0,z\rfloor \right)}R\left(x_{2},y_{2}\right)\right)\times AR \end{array}$$

Note: the function *R*(*x*, *y*) returns a random value following a uniform distribution in the range (*x*, *y*).

### 2.2. Parameter Setting

Before the simulations can be performed, several internal parameters have to be set in this model. According to the model mentioned above, the tumor size will eventually grow if *DR* > *AR*. To accommodate faster computational times, we choose an initial *AR* = 0.05 in this model, which means 5% of cells would die in each iteration. The value *DR* = 0.1 is then used for simulating precancerous cells with enough selective advantage, and the *DR* values for cancer cells are set in the range of 0.15 to 0.2. The maximum initial tumor size *n* is set to 100, and the mutation rate *MR* is set to 100, in reference to a previous study [22].

Driver mutations contribute a fitness advantage to cells. However, the strength of this advantage varies greatly. Bozic et al. [23] suggest a driver gene will bring a 0.4% fitness increase, while McFarlanda et al. [24] believe that this value is in the range of 10 to 60%. According to Kamel’s study, the fitness advantage brought by the driver gene may vary from 0.1 to over a hundred folds [25], which is due to the intrinsic difference of the cells. Thus, we believe that the fitness increase from a driver gene should be a more stable value, which is also easily explained from a biological point of view. In addition, to simplify the problem, only those strong drivers are recognized as “drivers”, insufficient fitness advantages brought by mini drivers are combined into random effects of passengers. In this model, we apply a dichotomous scheme to discriminate drivers from passengers. A “driver event” will lead to a new subpopulation, together with a *DR* increase of 0.05~0.15, an *AR* decrease of 5%, and an *MR* increase of 10~20 based on its originating source subpopulation. For hallmarks of specific types, a greater change in one of the parameters is used instead (0.1~0.25 for *DR*, 40% for *AR*, and 50~100 for *MR*, respectively). The different alterations correspond to the inclinations towards (1) proliferation and survival, (2) invasion and microenvironmental regulation, and (3) genomic mutability. This classification method of hallmarks is merely from the perspective of cell fitness, as biologically distinct traits may result in similar outcomes.

A chance of *MR*/6000 is used for the driver event occurrence. For each turn, random numbers of passenger mutations will be generated with a mean of *MR*/100, considering the greater number of passenger genes. For each subpopulation, all of the mutations are accumulated to calculate the tumor mutation burden (TMB). Unlike previous models, the probability of generating a new driver mutation from a subpopulation is only related to its *MR*, and not to the number of cells. The “Peto’s Paradox” phenomenon has been identified in previous experiments [26], namely that increased cell amount does not correlate with higher chances of developing cancer-driving somatic mutation. Considering the competitive pressure and absolute number differences in different spatial locations, cells carrying more driver mutations may not always have a competitive advantage. Besides, mutations could also be corrected by DNA repair processes [27]. We believe that in the context of competition, tumor cells are affected by an Allee effect [28]. In our model, *MR* may randomly increase or decrease at each iteration. However, a 1/10 probability of *MR* changes is in logarithmic proportion to the subpopulation size, which means *MR* of larger subpopulations may increase faster, while *MR* of small-sized (less than 10) subpopulations are likely to decrease.

According to McFarlanda et al. [24], passenger mutations have deleterious effects (~100 times weaker than driver mutations) to cancer cells. We decided to accept this theory, as the hypermutator tumors would contain more driver events, but their clinical phenotypes were similar to others. The values *x*_1_ and *y*_1_ in Equation (2) are set to −0.001 and 0.0008, with a slight tendency towards decreased fitness. The values *x*_2_ and *y*_2_ in Equation (3) are set to −0.002 and 0.002, and the *z* value in these equations is set to 0.1. As a more lenient limit does not affect the overall tendency but slows down the computation significantly, the constant *c* is set to 10,000, enabling rapidly replicated simulations.

### 2.3. Therapies in the Model

According to the clonal evolution theory, intra-tumor heterogeneity changes in response to external pressures, which is reflected incisively in clinical treatment. To better describe the dynamic intra-tumor heterogeneity changes, we integrated therapies into this model as well. Cancer therapies may be roughly broken down into two categories, targeted therapy and general therapy. The general therapy aims at clinical phenotypes, such as tumor size, position, and stage. This type of therapy is guaranteed to harm the tumor regardless of the subpopulation. The other type is more diverse in implementations, and is often subclone-specific. In practice, the therapy might be cytostatic or cytotoxic, the former inhibits tumor growth, while the latter can kill tumor cells [29]. For a better illustration of rapid drug response, all therapies in this model are designated to be cytotoxic.

In this model, the targeted therapy is designed to randomly select a driver event in the most prevalent subpopulation, and harm all subclones carrying that event with high efficiency. For each iteration, the targeted therapy will cause a 25~60% decrease in subpopulation size per iteration, and temporarily lower the *DR* value. As for the general therapy, all subpopulations will have a 5~15% decrease with a random loss of less than 100, and a small *MR* increase, as a side-effect [30]. Considering the potential damage to the patient’s health, for simulations performed in this paper, the maximum duration for both therapies is limited to 10 iterations.

As specific cases of target therapies, surgery and immunotherapy is also incorporated. Surgery will cause an instant removal of the main tumor mass, and only cells that have metastasized/invaded other tissues may survive from the surgery. If the “genome instability and mutation” hallmark is activated in a tumor subpopulation of size larger than 100, up to 40% of its cells may metastasize, and the proportion of invasive cells is calculated based on a constraint ($e^{\left(n\times 10^{-9}\right)}$), in which *n* is the tumor size. Immunotherapy will harm the tumor cells with an efficiency positively correlated with their tumor mutation burdens. The efficiency is calculated by $e^{\left(0.02\times TMB\right)}$, with an upper limit of ~45% decrease per iteration.

## 3. Results

### 3.1. Examples of DynamicIntra-Tumor Heterogeneity Progression

For cancers of different conditions, their routes of evolution are likely to vary widely. To allow a convenient illustration of the dynamic changes of intra-tumor heterogeneity, we provide a visualization of the model. Several simulations using different initial attributes were performed, and we randomly selected some cases to demonstrate typical examples of the dynamic changes in tumor subpopulations.

At first, we compared the differences of dynamic heterogeneity with and without interclonal competitions. We modified the constraints to attenuate the competition among subpopulations and simulate a duplication process where cells do not die from competitions (Figure 2A). For comparison, simulation with the normal parameters is shown in Figure 2B. Apparently, the latter situation is in more accordance with the clonal evolution theory. According to Figure 2B, the subpopulation with higher fitness is likely to become the majority, however, after the tumor has grown mature, it may take a very long time for a new subpopulation to grow (Figure 2C). An opposite example is shown in Figure 2D, when the occurrence of the new subpopulation brings a new hallmark such as metastasis or angiogenesis, the limitations are loosened and the new subpopulation rapidly gained its predominance. Also, it is observable that the tumor heterogeneity may decrease with the progression of a new subpopulation. Although intra-tumor heterogeneity tends to be more complex over time, it may also decrease due to the constant competition among tumor cells or an altered environment. This result is in concordance with the selective sweep theory [31].

Next, we incorporated the treatment effects into the simulation. The case in Figure 2C has been used as a basis to perform therapies. The targeted therapy has successfully eliminated the major subpopulation. However, a more malignant subpopulation rapidly proliferated (Appendix A). Due to the environmental restriction and the chronological order of occurrence, a subpopulation with higher fitness might only form a small proportion of the tumor mass. There is a possibility that the targeted therapy will worsen the condition. This phenomenon has been observed in previous studies [32]. If we start a targeted therapy before the emergence of the new subpopulation (Appendix A), the tumor-free time could be longer, but with only one course of treatment, a recurrence seems still inevitable. Then, after the first therapy, two more general therapies were applied, and the tumor size has been controlled for a long time period (Appendix A). Note that the downward trend of tumor size continues unabated after the second course, which is due to the accumulated negative fitness by numerous passenger events as the mutation rate is elevated by the general therapy’s side effect. This result follows a process known as Muller’s ratchet [33].

### 3.2. The CES Webserver

An online application based on this model has been developed to help users quickly simulate the dynamics of tumor heterogeneity or to explore the impact of specific factors on treatment responses. All of the simulations are performed in R. With the input dataset of initial attributes of tumor subpopulations, the underlying simulation algorithm will generate an R data.frame for tracing back the whole tumor growth process and for better illustration. The subpopulations are tagged to record the acquired driver events, including the initial “driver mutation” and newly acquired cancer hallmarks. The streamgraph (https://github.com/hrbrmstr/streamgraph, accessed on 17 February 2022) package are used to visualize the dynamics of tumor subclonal progression. To generate the input dataset, users may manually set the *DR*, *AR*, *MR*, and tumor size *n* of up to three initial subpopulations. The webserver is built under the Shiny framework using flexdashboard (https://pkgs.rstudio.com/flexdashboard/, accessed on 17 February 2022).

Figure 3 shows a screenshot of the CES webserver. The CES webserver consists of two main pages and a help page. On the “Clonal Evolution” page, users can modify various initial attributes through the left panel. Note that, the same input attributes may produce widely varying results due to randomness. It is highly recommended to repeat the simulation with same attributes, to check for the stochastic and uncontrollable nature of tumor progression. Click the “Start Simulation” button in the left panel to run the simulation with the currently set attributes. A stream plot indicating the dynamics in the size of tumor subpopulations will appear on the right side, and some other statistics are shown underneath. The lower three tabs show the number of tumor subpopulations, their emergence time and activated functions, as well as the two types of metrics we have defined (described in Section 3.4). After the simulation is completed on the current page, the user can enter the “Therapy simulation” page, and select the desired therapies and their start time points. Users may adjust the durations of the therapies. For the general therapy, the upper limit is 10 iterations; for the immunotherapy and targeted therapy, the limit is 15 iterations. The duration parameter does not affect the behavior of the surgery. Time interval between all treatments is not less than 10 iterations. Users can apply treatments at different time points on the results of the current simulation to observe the effect of a specific treatment on the tumor. Again, due to the random nature of target selection, it is recommended to try the simulation repeatedly to get an approximate idea of the potential impacts of therapies. Two panels below the main plot provide information showing the dynamics of the sizes and the TMB of subclones. The “Help” page provides basic guidance and recommendations to use the tool.

With the aid of this web application, users may explore possible associations among the subpopulation dynamics, the treatment outcome and other independent variables.

### 3.3. Simulation Results

Due to the stochastic nature of the model, simulations with the same input could lead to the vast differences in tumor progression. Therefore, a certain number of replications is needed to reveal the dynamic patterns of tumor progression and investigate the impact of individual parameters. We have performed 200 replicated simulations for each set of different initial parameters in order to assess their effects on results. One parameter is altered at a time, on the basis of an initial setting (*DR* = 0.2 for tumor cells/0.15 for precancerous cells, *AR* = 0.05, *MR* = 100, tumor size = 30, and 1 type of initiating subclones). First, we performed simulations for different *DR*s, *MR*s, and initial numbers of subpopulations. The final (after 200 iterations) tumor size is shown in Appendix A. The peaks in the plot indicate natural barriers restricting tumor growth. Mutations affecting specific factors/pathways are often required to ablate these barriers by activating new metabolic strategies or introducing new vessels, etc. Possibly due to competitive pressure, the increase in the number of initial subpopulations did not result in an increase in tumor size, which also suggests that intra-tumor heterogeneity is difficult to identify by phenotype. To be more detailed, the simulations for the whole progression process are illustrated by boxplots. As shown in Appendix A, a higher number of tumor subpopulations brings higher randomness in the development process. High levels of fluctuation at earlier times indicate a Strong Selection Weak Mutation (SSWM) evolutionary regime [34]. Likewise, the impact of randomness is greater when the fitness is less sufficient. Appendix A demonstrates the simulations with different mutation rates. Tumors may grow with similar efficiency with a range of mutation rates. However, a high mutation rate (*MR* = 500) may lead to a reduction in total size during later tumor development. When the mutation rate is too high (*MR* = 1000), a rapid decrease of tumor size starts early, implying a heavy mutation load can be a burden to the tumor.

Next, we explore the differences in treatment response across different initial conditions. Appendix A show simulations for a single targeted therapy, targeted therapy followed by general therapy, and two targeted therapies, respectively. All therapies start at time 100, while the initiating tumor size is set as 100 to ensure tumors to be well-grown. As can be seen, the effect of targeted therapy is more pronounced for tumors that start with only one type of subclones, as they tend not to have developed more subpopulations in sufficient amounts. Targeted therapy followed by general therapy can effectively stop tumor development, but two targeted therapies often cannot cover all tumor cell subpopulations, leading to the development of recurrence.

Subsequently, we explore the impact of different treatment time points on treatment outcomes. We performed targeted therapy followed by an adjuvant general therapy, based on the simulated data of the two starting subpopulations described above. The time to start therapies is set to 100. We simulated the time interval of 5, 10, 15, and 20, respectively, and counted the tumor size at time 170 for comparison with that in time 100. Figure 4A demonstrates the difference in outcomes between the four therapeutic strategies. We can see that the treatment effect gets progressively worse as the treatment interval increases, with 5 and 10 being the best time intervals. According to previous studies, patients who commenced timely adjuvant therapy had the best survival outcomes [35], delay beyond a certain time is associated with compromised survival [36]. We then explored the difference between adjuvant and neoadjuvant therapies. Based on the same data, we chose surgery as the main therapy, the general therapy as the adjuvant/neoadjuvant therapy (that is, after/before the main therapy). The first treatment is performed at time 100, and the second at time 120. Strikingly, the difference in results was very significant (Figure 4B). We suppose that, at a specific state, an earlier general therapy may repress minor subpopulations, thus inducing a heterogeneity decrease, or eliminating micro-metastases, to keep all remaining tumor cells at the surgery location. For comparison, a delayed general therapy after surgery gives these minor subpopulations more time and space to grow and spread, while a delayed surgery after general therapy does not provide discernible advantage for the main tumor mass in confronting a surgery. Similar results are found in previous studies [37,38].

### 3.4. Intra-Tumor Heterogeneity and Treatment Responses

To assess intra-tumor heterogeneity, we defined two categories of metrics. The “dominant clone proportion” indicates an absolute heterogeneity which is defined as the proportion of the predominant cell subpopulation, while “off-target probability” indicates a potential heterogeneity which is the proportion of driver mutations that are not contained in the predominant cell subpopulation to all driver mutations.

Having defined these two metrics, we then investigated if their values can affect the treatment outcome. We chose the previously described “three initial subpopulations” group of simulations as the input data. Targeted therapies (10 iterations) were performed at the time 100, and the fold-changes of the tumor size after 50 iterations were calculated. A moderate correlation was found between the two metrics at time 100 and the final fold-changes (Figure 5). According to the simulations, a lower proportion of subclonal cells or subclonal driver mutations is likely to lead to a better result of the targeted therapy.

We have also simulated the situation where only a single subpopulation of cancer cells is initially present at the beginning of tumor progression. The simulations show that absolute heterogeneity is still associated with treatment efficacy, but the number of new driver mutations is no longer a critical factor (Appendix A). Due to the first-mover advantage, the earliest cells containing a driver mutation tend to dominate, while the rest of the cells are their descendants and can be equally targeted by the treatment.

Next, we investigated the trends of these two types of metrics over the course of cancer development. With the same input dataset, results have shown that both metrics tended to have a parabolic shape, due to the turnover of the dominant subpopulation within the tumor mass (Figure 6). The heterogeneity may decrease after the mid-term due to the competition between subpopulations, which may suggest a chance of better outcome from a delayed cancer treatment. However, since tumors tend to spread or metastasize in the late stages of development, spatial extensions can weaken the intensity of direct competition and thus increase heterogeneity. The proper use of neoadjuvant therapy is also recommended to reduce heterogeneity before proceeding to primary cancer treatment [39].

### 3.5. Simulations of Tumorigenesis

The previous simulations were started directly at a tumoral state, to explore the process of transition between normal cells and tumor cells and to match the model to reality, we then start the simulation at the normal cell state and investigate the important factors in this process. To make the changes more refined, we lowered the *AR* and *DR* values for each iteration, and calculated the times reaching a detectable size for different input data. Based on the tumor size distribution of simulations in Appendix A and previous studies [40,41], we set the detectable tumor size to 10,000. On a basis of *AR* = 0.02, initial tumor size = 30 and 1 single subpopulation, we applied different *MR* and *DR* values to check for their importance. The initial attributes and results are shown in Figure 7. According to the studies on the proliferating cell nuclear antigen index [42] and cell division rates [25], for solid tumors such as glioblastoma multiforme and colorectal cancers, the iteration interval T is approximately 8~10 days when *AR* = 0.02. We select T = 10 days to estimate the real-time. With no initial fitness advantages, the time to develop a detectable tumor falls in a range of 3.3–22.5 years, with a median of 9.6 years. This result is more consistent with the results of the previous mutation study [23] (~8.3 years for a new driver mutation to occur), but deviates from another estimation [43] (~17 years to develop an advanced cancer). Potential reasons could be: first, compared with the fixed minimum detectable size, the sizes of tumors in that study were significantly larger, and second, the division frequency of normal cells was not as frequent as cancer cells, so in the early period, the time interval needs to be lengthened. As shown in the figure, compared to *DR*, increased *MR* significantly advanced the time of tumor development, implying the danger of environmental factors that increase gene mutations.

## 4. Conclusions and Discussion

To investigate the temporal variability of intra-tumor heterogeneity and its relation with treatment response, we constructed a simulation model based on the clonal evolution theory, which has been validated by the enormous amount of evidence [44]. Other theories [45,46] are not considered as the theoretical basis, as their evidence could also be explained under the frame of clonal evolution. Various important factors are involved in this model, including driver/passenger mutation events, environmental limitations, diverse hallmarks, stochastic disturbance, and different types of therapies. With only three main attributes, the complexity of tumor evolution was drastically reduced. The results have shown that, using a small number of parameters, this model is capable to prefigure the subclonal evolution dynamics to reflect the reality to a certain extent. Although the model is designed to simulate a trend, not real situations, it still may work as a framework. Through its simplicity, one may focus on the research question towards specific factors of interest. Several factors relating to complex characteristics are not considered in this study, including the effect by the tumor microenvironment and germline variations. Introducing too many parameters raises the probability of increasing the cumulative bias. A more precise simulation may be obtained by further improvement of model structure and parameter adjustments.

Various types of models have been developed to depict tumor evolution in past years. Models of two main categories [47] have been proposed. One is the Moran process, wherein the population size remains the same and only the types are being substituted, and one is the branching process, wherein the cells proliferate by generations. Our model is similar to the latter one (i.e., without a “generation” where all cells divide or die) but instead use probabilities to simulate the action of individual cells. In addition, many models simulate the growth of cancer cells as an exponential increase process. In practice, this phenomenon mostly occurs in early tumor stages, where too many limitations in the organism tend to lead to a more subdued growth in tumor size. The model proposed in this paper both demonstrates the limitation and offers the possibility to break through it.

Some of the observations from the simulations may have clinical significance. First, the proper combination of general therapies and other therapies can lead to a better outcome. Although a general thought is that compared to clinical phenotypes, molecular signatures are more suitable therapy targets. Actually, only a small portion of driver mutations are practically targetable. Also, when the intra-tumor heterogeneity is high, a sole targeted therapy may not inhibit tumor growth. With an accurate estimation of the heterogeneity landscape, an appropriate therapeutic strategy may maximize the treatment effect. Second, the different metrics of intra-tumor heterogeneity can be correlated to the treatment outcome. The performance of targeted therapy is correlated to the proportions of target cells [48], and the recurrence of cancers is confirmed to be related to intra-tumor heterogeneity [49]. To perform the most effective therapy, the identification of intra-tumor heterogeneity is common in the clinic [50]. With the selective pressure exerted on tumor cells, the distribution of cell subpopulations may change, a therapy/immuno-resistant minority subpopulation may eventually dominate. The dynamics of intra-tumor heterogeneity should be actively tracked to timely adjust the therapy. Finally, we have also observed a selective sweep phenomenon in this model. The invasive nature of tumors makes this situation less common in practice, as there are always dissociated tumor cells apart from the tumor mass. However, in particular situations, the heterogeneity decreases with tumor progression, a delayed treatment timing might lead to a better result. Theoretically, to reduce heterogeneity, one may exert directional selective pressure on tumor cells before treatment, but currently no mature practice is available. We hope all of these investigations and observations in this paper may contribute to improving the clinical practice.

## Figures and Tables

**Figure 1 cancers-14-01645-f001:**
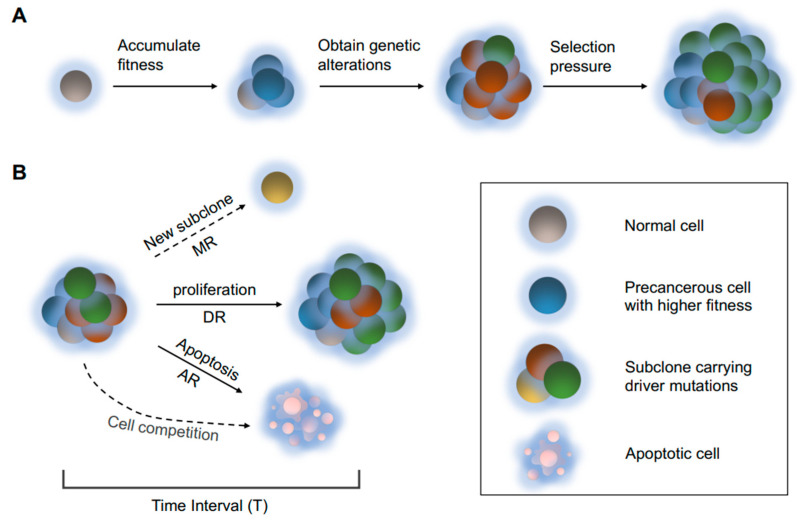
(**A**) An illustration of the clonal evolution process. Cells may accumulate mutations with different functional consequences over generations. During the process, a “driver mutation” may occur and provide a substantial increase in cell fitness, leading to further expansion of the corresponding clone. Different subclones in tumor may yield distinct responses to external pressure. (**B**) A schematic of cell population alteration process in the model. In the time interval T, all subclones may proliferate or decease individually according to their division rates (*DR*s) and apoptosis rates (*AR*s). The chances for the emergence of a new subpopulation correlates to the mutation rate (*MR*). When the total cell population is high, cell competitions will cause the elimination of less competent cells.

**Figure 2 cancers-14-01645-f002:**
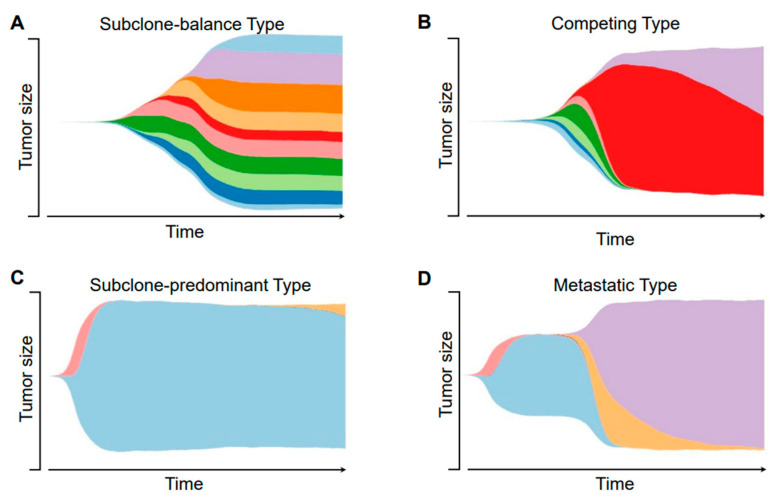
Representative illustrations of the dynamic intra-tumor heterogeneity. This figure shows four examples of the dynamic progression of different tumor subpopulations, each representing a different type. Different colors represent for different subclones, the height of the main body indicates the tumor size. (**A**) The subclone-balance type; (**B**) the competing type; (**C**) the subclone-predominant type; (**D**) the metastatic type.

**Figure 3 cancers-14-01645-f003:**
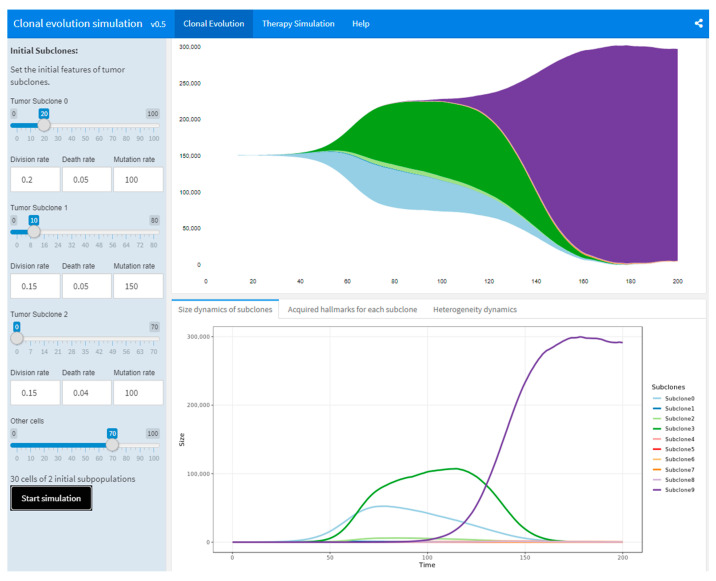
An online application for simulating the subclonal evolution of tumor cell populations. A screen-shot of the simulation output is shown. Initial values of the attributes for simulation are shown on the left panel. The streamplot indicates a dynamic progression of cell subpopulations. A mouse hovering over the main body shows the size of the corresponding subpopulation.

**Figure 4 cancers-14-01645-f004:**
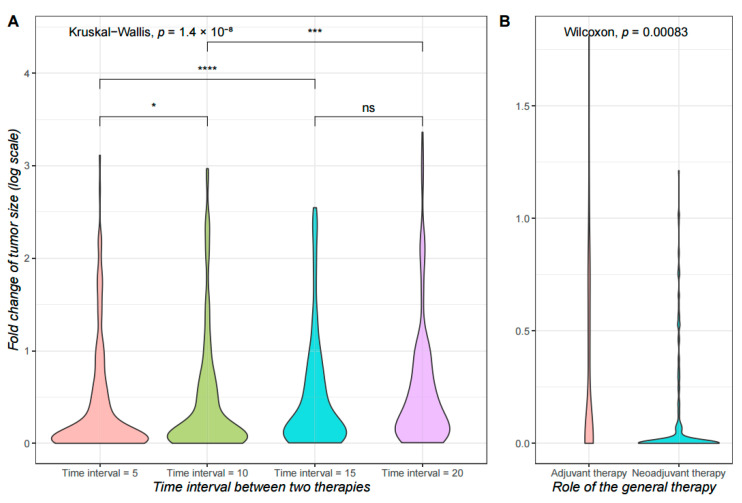
Treatment outcomes of different timings. The tumor size change across simulations is shown as violin plots. The y axis indicates the log-scaled fold-change of tumor size after 50 iterations of simulations. For all simulations, the first therapy is performed at time 100, and the fold-change is calculated between tumor sizes at time 100 and time 170. Simulations for each condition is repeated 200 times. Significance levels of difference among groups are shown (*: *p*-value ≤ 0.05; ***: *p*-value ≤ 0.001; ****: *p*-value ≤ 0.0001; ns: *p*-value > 0.05). (**A**) The *x* axis indicates whether the general therapy functioned as adjuvant therapy to the surgery (after surgery), or as a neoadjuvant therapy (before surgery); (**B**) The *x* axis indicates different time gaps between the first therapy and the adjuvant therapy.

**Figure 5 cancers-14-01645-f005:**
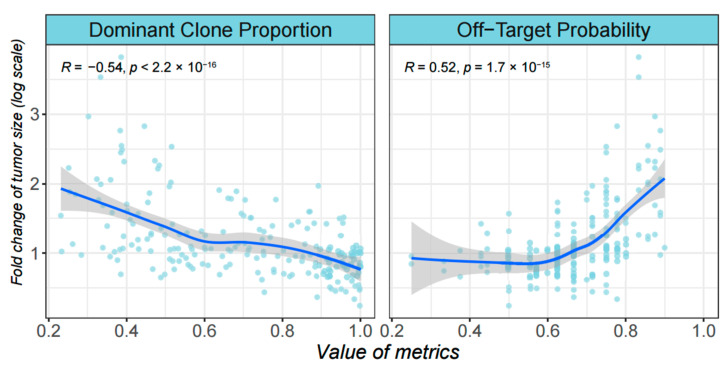
Correlations between the treatment outcome and the ITH metrics. Each replicated simulation is represented by dots on the scatterplot, where the *x* axis denotes the values of the two metrics when the therapy begins, and the *y* axis denotes log-transformed fold change of tumor size at 50 iterations after the therapy is over.

**Figure 6 cancers-14-01645-f006:**
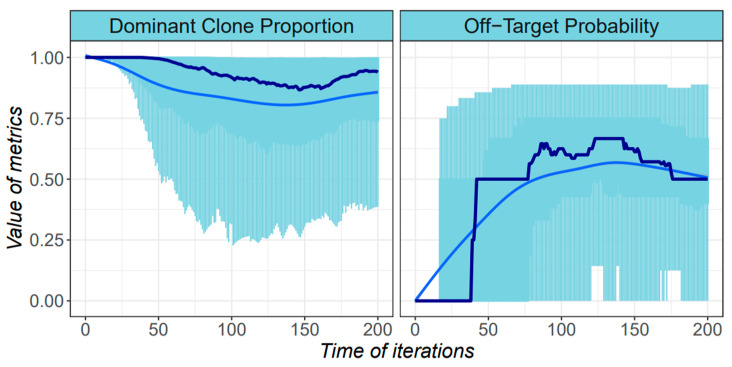
The distribution of the ITH metrics among simulations. Boxplots are used to represent the distributions of corresponding values from all of the simulations. The *x* axis denotes the time, and the *y* axis denotes the value. 200 simulations are performed and two metrics are calculated at each time point. The dark blue line indicates the median value from all simulations, and the blue curve indicates a loess regression based on the values.

**Figure 7 cancers-14-01645-f007:**
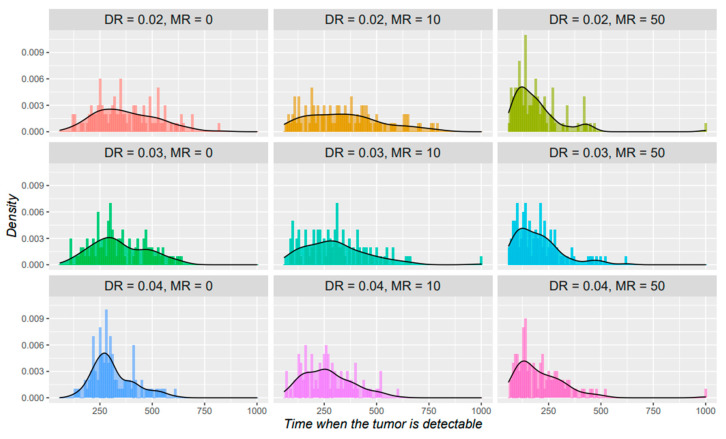
Time distribution when tumor reaches a detectable size. The *x* axis indicates the time when the tumor size reaches 10,000, the *y* axis indicates the density for each binned time periods. Each panel represents for a different initial setting of the parameters.

**Table 1 cancers-14-01645-t001:** Parameters in the model.

Parameter Full Name	Abbreviation	Denotation	Default Values in the Model
Division rate	*DR*	The duplication rate of tumor cells in a time interval *T*.	0.2 for cancer cells, 0.15 for precancerous cells
Apoptosis rate	*AR*	The apoptosis rate of tumor cells in a time interval *T*.	0.05
Mutation rate	*MR*	The count of expected mutations occurring in a time interval *T*.	100

## Data Availability

The data presented in this study are available on request from the corresponding author.

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
