# Peer review of "Simulating the Dynamic Intra-Tumor Heterogeneity and Therapeutic Responses"

_cancers, 2022, doi:10.3390/cancers14071645_

Round 1

Reviewer 1 Report

The paper presents a simple but illustrative model of clonal evolution resulting in tumor heterogeneity. It enables to estimate the effect of treatment in simple situations. An interactive web-platform allows to “play” with the model in different settings.

I like this paper. Clonal evolution is a key for understanding cancer. Overall its complicated and not understood in detail. The model provided breaks this down to a relatively simple scenario which helps learn about the basic mechanisms of tumor development.

I suggest the following points for improving the paper:

  1. The text descriing the model is okay but insufficient to catch the basic ngredients. Might the authors want to add a table which lists the model parameters, their abbreviations and meaning, as well as basic settings.
  2. Might the authors want to add a figure schematically illustrating the mathematical model and its ingredients, e.g. by considering Fig. 1 or designing a separate figure.
  3. Can you please explain the meaning of the exponential term in Eq.(1).
  4. (1) is redundant (144-145)
  5. 4: please assign Y-axis.
  6. 5: x-axis, value of what, please annotate more precisely....the same for FC (of what, Y-axis)....Also fig. 6: Ratio of what, what time...etc...Fig. 7: count--> cell count
  7. Conclusions and discussion...reverse order: Discussion and conclusions...

Author Response

Reviewer: 1

Comments to the Author

The paper presents a simple but illustrative model of clonal evolution resulting in tumor heterogeneity. It enables to estimate the effect of treatment in simple situations. An interactive web-platform allows to “play” with the model in different settings.

I like this paper. Clonal evolution is a key for understanding cancer. Overall its complicated and not understood in detail. The model provided breaks this down to a relatively simple scenario which helps learn about the basic mechanisms of tumor development.

I suggest the following points for improving the paper:

  1. The text descriing the model is okay but insufficient to catch the basic ngredients. Might the authors want to add a table which lists the model parameters, their abbreviations and meaning, as well as basic settings.

Response: Thank you for this suggestion. A table containing the model parameters, the abbreviations, meanings and basic settings are added.

  1. Might the authors want to add a figure schematically illustrating the mathematical model and its ingredients, e.g. by considering Fig. 1 or designing a separate figure.

Response: Thank you for this suggestion. We have modified Figure 1 to include a schematic illustration of the model.

  1. Can you please explain the meaning of the exponential term in Eq.(1).

Response: Thank you for asking. This exponential term is a constraint between 0 and 1, which will decrease as the tumor size goes up. It simulates the environmental limitations to dynamically adjust tumor growth. When the tumor size N is low, this term approximates to 1, thus the cells can proliferate without limitations (in an exponential manner). As N increases to a certain level, a dynamic equilibrium between tumor growth and environmental constraints will be reached. Notably, changes in DR and AR may alter the final value of this equilibrium. In this study, driver mutation events were classified into three major categories, which differ in their effects on DR and AR, and consequently on the final value of the equilibrium. In short, this term makes use of available parameters to simulate a Gompertzian growth of tumor, while distinguishing the differences in the consequences of different mutations.

  1. (1) is redundant (144-145)

Response: Thank you for bringing this point. We have now corrected the errors in formatting the equations.

  1. 4: please assign Y-axis.

Response: We have altered the title of y axis for Figure 4.

  1. 5: x-axis, value of what, please annotate more precisely....the same for FC (of what, Y-axis)....Also fig. 6: Ratio of what, what time...etc...Fig. 7: count--> cell count

Response: Thank you for these comments. The “Count” in Figure 7 means the count of simulation results felling in each bin. To avoid misunderstanding, we change the x-axis title to “Density”. All these figures have been revised as per your advice.

Conclusions and discussion...reverse order: Discussion and conclusions...

Response: Thank you for your carefulness. The title of this section has been revised as “Discussion and conclusions”.

Reviewer 2 Report

The authors provide overall good summary of the tumor heterogeneity landscape. Some of the general comments: could they comment on the metastasis aspects? Does the model apply to 2ndary tumor as well? From a general cancer perspective, will different mutations (in tumor suppressor or oncogene/driver) affect MR differentially? How genetic information like somatic germline variants would be incorporated into the model.

Author Response

Reviewers' Comments to Author:

Reviewer: 2

Comments and Suggestions for Authors

The authors provide overall good summary of the tumor heterogeneity landscape. Some of the general comments: could they comment on the metastasis aspects?

Response: Thanks for this comment. In this paper we have referred to tumor metastasis and set the conditions for its occurrence. Depending on the various tumor types and metastasis sites, environmental change may bring different selection pressures, which in turn make cancer cells in the metastases and the primary site exhibit different characteristics. However, this encompasses too many complexities, and our focus is just to simulate the impact of metastatic cells on surgery treatment. As long the subclones are with the same genomic composition, we consider them as a whole, despite the spatial discontinuity. Some related descriptions have been added in the revised manuscript.

Does the model apply to 2ndary tumor as well?

Response: The model can also be applied to simulate a secondary tumor. Actually, most of the simulations in this paper are for cells that have accumulated a certain level of fitness, or cancer cells. Simulations of secondary tumors only require appropriate adjustment of parameters, and one way to do this is to use the parameters obtained after a number of iterations as initial values.

From a general cancer perspective, will different mutations (in tumor suppressor or oncogene/driver) affect MR differentially?

Response: This is a good question. In this paper, we divided the driver mutations roughly into three categories. As tumor suppressor or oncogene may correspond to a same function, it is perhaps more accurate to classify them according to their functional consequences rather than according to their own roles. Of course, in practice, the mutations that affect MR are essentially from DNA damage response pathways, a significant fraction of which are tumor suppressors, so that alterations of tumor suppressors are indeed more likely to increase MR in cancer subclones. However, this point is integrated into the stochastic part of the model and is not directly / explicitly reflected.

How genetic information like somatic germline variants would be incorporated into the model.

Response: Thanks for this comment. To make things easier, currently the model does not provide a direct option for germline variants. Depending on the consequences of different germline variants, one may be able to adjust the initial parameters to simulate them. For example, some high-risk germline variants tend to result in easier emergence of certain driver events, and a moderate increase in MR can mimic some of their properties. We hope you find these answers and revisions acceptable.

Reviewer 3 Report

The authors present an evolutionary model for cancer to simulate tumor evolution and impact of both intra-tumor heterogeneity and treatments on the course of the disease. The analysis is nicely conducted, and the authors made the tools accessible and easy to use which is very much appreciated. The work is interesting but needs to be refined to better link the in-silico analysis to the actual clinical/biology reality of cancers. This effort is necessary to derive new sound knowledge from the model and make it more understandable to researchers. Please find below my comments through the sections of the manuscript.

  1. General remarks
    • The model is well explained but the choice of the parameters should be better justified or at least compared to what is expected in reality: what are the expected mutation/duplication rates of cancer cells for instance?
    • The potential differences between cancer types are not discussed. Would the parameters need to be changed according to cancer types (liquid/solid) or cancer subtypes (in breast cancer for instance: triple negative, or ER+ breast cancers)?
    • Potential enhancements of the model should be further developed. For instance, the role of the tumor microenvironment, and especially the immune microenvironment is not included in the model. Maybe a point to address in the discussion.
    • A section dedicated to the validation of the model would help the analysis gain credibility. How reinsuring the simulations are compared to what is observed in reality?
    • Regarding open science:
      • The tool is easily accessible online and easy to use.
      • Some guidelines on potential scenarios or examples of values for the different parameters could help the user to dive into the possibility of the tool.
      • Will the code be made available as well? This would be a plus.
    • Supplementary data:
      • Legend should be more detailed, within the supplementary document itself.
  1. Introduction:
    • Line 73: “In many situations, the comprehensive genomic profiles can be reduced to profiles of a few driver genes.” This is not entirely true considering the importance of markers such as tumor mutational burden, or mutational and rearrangement signatures that use the genome wide alteration landscape.
    • Line 74: Does “Subtypes” refer to subclone? If yes it would be clearer to keep subclone, if not, the concept should be defined.
  2. Point by point remarks
    • Section 2.2 parameter setting
      • Ref 22: the reference seems to have been performed out of the context of cancer. How relevant are these rates when dealing with tumor cells?
    • Results – 3.1
      • Line 227: can you better explain how you define an “in vivo” situation and a “laboratory” situation?
    • Results - Section 3.3
      • Line 320: MR=1000 is written in the text but not shown in figure S4, could you add this panel?
      • Line 321: In some cancers the tumor mutational burden could be a biomarker for sensitivity to immunotherapy (https://www.annalsofoncology.org/article/S0923-7534(19)30997-4/fulltext ). Would it be possible to evaluate this marker based on the mutation rate given to the simulation?
      • Line 325: the time is set to 100 to initiate the treatment at a time where the tumor is big enough. What is the median/average size of the tumor at that time? Is this comparable with a clinical scenario?
    • Results – section 3.4
      • Figure 5: the name of the variables in the plot should appear also in the text to help the reader.
        • For “off-target probability”: is this the second heterogeneity metric? Maybe formulate it more explicitly.
        • Figure 6: the legend mentions boxplots but there is no boxplot on the figure. The legend should define more precisely the y axis.
      • Discussion
        • Line 465: “the heterogeneity decreases with tumor progression, a delayed treatment timing might lead to a better result”: this principle seems to be difficult to apply in the clinic for ethical reason. Indeed, it seems difficult to not treat a patient who is progressing. The applicability in the clinic should be discussed here.
      • Supplementary figures:
        • Figure S2: a y axis should be added.
  1. Minor remarks:
    • Line 54: “Hundreds of recent researches have evidenced”, remove hundreds or add more references
    • Line 214: “patient’s health, the maximum duration for both therapies is limited to 10 iterations” targeted therapies are supposed to be better tolerated than systemic treatments, would it be possible to adapt the number of iterations for each therapy type?
    • Line 219: can you develop a bit more on the definition of the constraint?

Author Response

Reviewers' Comments to Author:

Reviewer: 3

Comments and Suggestions for Authors

The authors present an evolutionary model for cancer to simulate tumor evolution and impact of both intra-tumor heterogeneity and treatments on the course of the disease. The analysis is nicely conducted, and the authors made the tools accessible and easy to use which is very much appreciated. The work is interesting but needs to be refined to better link the in-silico analysis to the actual clinical/biology reality of cancers. This effort is necessary to derive new sound knowledge from the model and make it more understandable to researchers. Please find below my comments through the sections of the manuscript.

  1. General remarks

The model is well explained but the choice of the parameters should be better justified or at least compared to what is expected in reality: what are the expected mutation/duplication rates of cancer cells for instance?

Response: Thank you for this valuable suggestion. We have provided a table to describe the model parameters, the abbreviations, meanings and basic settings in the revised manuscript. Specifically, to answer the question, the expected mutation / duplication rates differ for different subtypes / stages / originating tissues of tumors. Some publications (e.g. PMID: 22820252, PMID: 26785814) have provided data of these rates, however contradictory results may be presented, possibly due to incomplete collection of samples. In this paper, we have provided an example matching the parameters to reality in Section 3.5. Based on this example, users may adjust the parameters according to their interested cancer types. As per your advice, we have added a separate page in the web application, containing instructions, recommended initial values and other useful information.

The potential differences between cancer types are not discussed. Would the parameters need to be changed according to cancer types (liquid/solid) or cancer subtypes (in breast cancer for instance: triple negative, or ER+ breast cancers)?

Response: Thank you for this point. The optimal parameters will definitely change for different cancer types. If possible, alterations should be made particularly according to the characteristics of the genomic events driving that subtype. For example, IDH1-mutated gliomas had a 2.5-fold slower duplication rate than IDH1-wild type gliomas (PMID: 27576872). Also, the model is designed for solid tumor. Due to the complicated spatial structures of liquid tumors, competitions of subclones can be indirect and hard to simulate. We have added this statement in Section 2.1.

Potential enhancements of the model should be further developed. For instance, the role of the tumor microenvironment, and especially the immune microenvironment is not included in the model. Maybe a point to address in the discussion.

Response: Thank you for this suggestion. The tumor microenvironment, especially the compositions of different immune cells, the pattern and degree of infiltration and other immunologic characteristics, are too complex for this model. Not to mention the difficulty of quantification, introducing too many parameters that can influence tumor development will only increase the cumulative bias and the results will become more unreliable. Simplification is all about trade-offs, some factors need to be neglected to better focus on interested research question. All our updates will be based on existing parameters. We have discussed this question in the Discussion and conclusions section.

A section dedicated to the validation of the model would help the analysis gain credibility. How reinsuring the simulations are compared to what is observed in reality?

Response: Thank you for this valuable suggestion. First, since the main outcome of the model is the dynamic size of different subclones, there is hardly any study able to provide such data. Talking about intra-tumor heterogeneity (ITH), researchers are more concerned with clinical-related issues such as survival. Sample-level ITH is provided by numerous studies, but as single values that is not dynamic and difficult to refine to a subclonal level. Second, the stochastic nature of the model makes "validation" less meaningful. In reality, patients also have distinct characteristics, a significant result might come from randomness, like a biased selection of the data. In this paper, the “Simulations of Tumorigenesis” section is an attempt for linking the model to reality. We believe, a more important point of the simplified model is to reflect an overall trend, focusing on particular features. We hope you are satisfied with our answer.

Regarding open science:

The tool is easily accessible online and easy to use.

Some guidelines on potential scenarios or examples of values for the different parameters could help the user to dive into the possibility of the tool.

Response: Thank you for this comment. A new page in the CES webserver has contained guiding information.

Will the code be made available as well? This would be a plus.

Response: Thank you for this suggestion. We're sorry, but we're still considering updates beyond the base functionality and don't plan to make the code public until we get to the final version.

Supplementary data: ""

Legend should be more detailed, within the supplementary document itself.

Response: Thank you for this suggestion. We have now uploaded a supplementary document containing the detailed supplementary legends.

  1. Introduction:

Line 73: “In many situations, the comprehensive genomic profiles can be reduced to profiles of a few driver genes.” This is not entirely true considering the importance of markers such as tumor mutational burden, or mutational and rearrangement signatures that use the genome wide alteration landscape.

Response: Thank you for this comment. There are indeed many other important markers, and in this paper we just focused on the aspect of driver mutations. On one hand, it is common to identify cancer subtypes by driver genes. The status of one or a few key genes often reflect the overall characteristics of the cancer. On the other hand, from a therapeutic point of view, considering the limitations of available therapies, there are often only a few genes of interest. We have changed the wording to “In some situations”, and added TMB as an observable value in the webserver.

Line 74: Does “Subtypes” refer to subclone? If yes it would be clearer to keep subclone, if not, the concept should be defined.

Response: Thank you for this point. The “subtypes” here refers to tumor molecular subtypes (e.g. basal or luminal breast cancer) which are distinguished by characteristics of few key genes. For most “subtype” terms throughout this paper, we are talking about different types of subclones. It seems that mixed use of “subclone”, “subtype” and “subpopulation” may cause some misunderstanding. Therefore, we have made it clear and rephrased some corresponding sentences. Besides, we have also checked other equivalent terms throughout the paper, and have made some alterations to ensure the consistency of terminology.

  1. Point by point remarks

Section 2.2 parameter setting

Ref 22: the reference seems to have been performed out of the context of cancer. How relevant are these rates when dealing with tumor cells?

Response: Thanks for your carefulness. This reference (as well as other references we checked) gave a basic estimation of genomic mutation rate. As for tumor cells, it is generally accepted that the mutation rate is higher, but there is still no consensus on a precise value, and the conclusions from different researches may vary by several orders of magnitude. There are also papers that refer to this value to model tumor mutations, so it can be considered as relevant. In addition, to ensure an effective accumulation of mutations, this paper does not consider DNA repair mechanisms. Despite the exact MR value in the model, it guarantees a higher actual number of mutations that would occur in the simulation from another perspective.

Results – 3.1

Line 227: can you better explain how you define an “in vivo” situation and a “laboratory” situation?

Response: Thank you very much for this comment. In the comparison of Figure 2A and Figure 2B, we are trying to illustrate competition between tumor subpopulations. What we are assuming is a more permissive developmental environment in which the tumor cells are not likely to die due to competition. Obviously, the term "laboratory" is not an accurate description. We have revised the wording and hope to receive your approval.

Results - Section 3.3

Line 320: MR=1000 is written in the text but not shown in figure S4, could you add this panel?

Response: Thank you for pointing out this, we are sorry for our negligence. We have updated the Figure S4.

Line 321: In some cancers the tumor mutational burden could be a biomarker for sensitivity to immunotherapy (https://www.annalsofoncology.org/article/S0923-7534(19)30997-4/fulltext ). Would it be possible to evaluate this marker based on the mutation rate given to the simulation?

Response: Thanks for the question. tumor mutational burden is thought to be related to the number of neoantigens on the cell surface, and its increase leads to improved immunotherapeutic efficacy. MR in the model is indirectly related to TMB, we can estimate TMB based on the random consequences by MR at each iteration. Inspired by this question, we decided to add TMB in this model, and add immunotherapy as an option for therapies.

Line 325: the time is set to 100 to initiate the treatment at a time where the tumor is big enough. What is the median/average size of the tumor at that time? Is this comparable with a clinical scenario?

Response: Thank you for this comment. Particularly for the dataset used, the range of tumor size is 30581 – 790706, with a median of 172531 and a mean of 209432, all of which larger than a detectable size. Converting to actual tumor volume, it should be 3 cm^3 to 79 cm^3, covering from early stage to late stage tumors. This data matched well with multiple cancer types despite the large variation in size across cancer types.

Results – section 3.4

Figure 5: the name of the variables in the plot should appear also in the text to help the reader.

For “off-target probability”: is this the second heterogeneity metric? Maybe formulate it more explicitly.

Response: Thanks for the comment. This is indeed the second heterogeneity metric, which we have mentioned in section 3.4, but the name is not explicitly mentioned. We have rephrased the corresponding paragraph, and added an equation for better demonstration.

Figure 6: the legend mentions boxplots but there is no boxplot on the figure. The legend should define more precisely the y axis.

Response: Thanks for the comment. The figure actually contains 200 side-to-side boxplots. For easier viewing, we integrated the images and text into one file, the original images in pdf format were converted to tiff format and details were lost in the conversion process. In this revision, all image files are also uploaded in pdf version, while the axis titles and legends are changed accordingly.

Discussion

Line 465: “the heterogeneity decreases with tumor progression, a delayed treatment timing might lead to a better result”: this principle seems to be difficult to apply in the clinic for ethical reason. Indeed, it seems difficult to not treat a patient who is progressing. The applicability in the clinic should be discussed here.  

Response: Thank you very much for this comment. We absolutely agree that not treating patients is inhumane. Actually, by “delay” here we do not mean an intentional delay, but simply point out the possibility of this situation. Given that the reason for this phenomenon is a reduction of intra-tumor heterogeneity during tumor development, it can be argued that early therapies that reduce heterogeneity or exert directed selection pressure may improve the outcome of main therapy. The corresponding discussion is in the Discussion and conclusions section of this paper.

Supplementary figures:

Figure S2: a y axis should be added.

Response: We have added a title for the ridge plot of Figure S2.

  1. Minor remarks:

Line 54: “Hundreds of recent researches have evidenced”, remove hundreds or add more references

Response: We have removed “hundreds”.

Line 214: “patient’s health, the maximum duration for both therapies is limited to 10 iterations” targeted therapies are supposed to be better tolerated than systemic treatments, would it be possible to adapt the number of iterations for each therapy type?

Response: Thank you for this useful suggestion. We have altered the maximum durations for different therapies, and made that as an option in the web application.

Line 219: can you develop a bit more on the definition of the constraint?

Response: Thanks for this comment. We have added a brief explanation for this part. Thank you again for taking the time to offer us your comments and insights related to the paper. We tried to be responsive to your concerns, and we hope you find these revisions rise to your expectations.

Round 2

Reviewer 3 Report

I thank the author for their work. They reply positively to all my comments.